# The Prediction and Prognosis of Fungal Infection in Lung Transplant Recipients—A Retrospective Cohort Study in South Korea

**DOI:** 10.3390/jof7080639

**Published:** 2021-08-06

**Authors:** Yae-Jee Baek, Yun-Suk Cho, Moo-Hyun Kim, Jong-Hoon Hyun, Yu-Jin Sohn, Song-Yee Kim, Su-Jin Jeong, Moo-Suk Park, Jin-Gu Lee, Hyo-Chae Paik

**Affiliations:** 1Division of Infectious Disease, Department of Internal Medicine, Severance Hospital, Yonsei University College of Medicine, Seoul 03722, Korea; stellangela.baek@yuhs.ac (Y.-J.B.); yunsukcC@yuhs.ac (Y.-S.C.); craphica@yuhs.ac (M.-H.K.); ayu870213@yuhs.ac (J.-H.H.); rubythyme@yuhs.ac (Y.-J.S.); 2Division of Pulmonology and Critical Care Medicine, Department of Internal Medicine, Severance Hospital, Yonsei University College of Medicine, Seoul 03722, Korea; dobie@yuhs.ac (S.-Y.K.); pms70@yuhs.ac (M.-S.P.); 3Division of Thoracic Surgery, Severance Hospital, Yonsei University College of Medicine, Seoul 03722, Korea; csjglee@yuhs.ac (J.-G.L.); hcpaik@yuhs.ac (H.-C.P.)

**Keywords:** fungal infection, lung transplantation, fungal-positive respiratory sample, *Aspergillus* spp., *Candida* spp.

## Abstract

(1) Background: Lung transplant recipients (LTRs) are at substantial risk of invasive fungal disease (IFD), although no consensus has been reached on the use of antifungal agents (AFAs) after lung transplantation (LTx). This study aimed to assess the risk factors and prognosis of fungal infection after LTx in a single tertiary center in South Korea. (2) Methods: The study population included all patients who underwent LTx between January 2012 and July 2019 at a tertiary hospital. It was a retrospective cohort study. Culture, bronchoscopy, and laboratory findings were reviewed during episodes of infection. (3) Results: Fungus-positive respiratory samples were predominant in the first 90 days and the overall cumulative incidence of *Candida* spp. was approximately three times higher than that of *Aspergillus* spp. In the setting of itraconazole administration for 6 months post-LTx, *C. glabrata* accounted for 36.5% of all *Candida*-positive respiratory samples. Underlying connective tissue disease-associated interstitial lung disease, use of AFAs before LTx, a longer length of hospital stay after LTx, and old age were associated with developing a fungal infection after LTx. IFD and fungal infection treatment failure significantly increased overall mortality. Host factors, antifungal drug resistance, and misdiagnosis of non-*Aspergillus* molds could attribute to the breakthrough fungal infections. (4) Conclusions: Careful bronchoscopy, prompt fungus culture, and appropriate use of antifungal therapies are recommended during the first year after LTx.

## 1. Introduction

Lung transplantation (LTx) emerged as a lifesaving treatment modality for end-stage lung disease with the first transplantation in 1963 [1]. The median overall survival duration in lung transplant recipients (LTRs) is 5.6 years, but this rate has increased to 7.9 years in LTRs who survive for over 1 year after LTx [2]. The 1-year survival rate after solid organ transplantation is usually influenced by two significant factors: allograft rejection and infection. Although immunosuppressive regimens have reduced the incidence of transplanted organ rejection, they confer a substantial risk of opportunistic infections, with invasive fungal infections (IFIs) being significant [3]. The common pathogens that cause IFIs after LTx are *Aspergillus* spp. (44%, most commonly *A. fumigatus*), *Candida* spp. (23%, most commonly *C. albicans*), and other molds [4]. Overall, fungal infection occurs in 15–35% of LTRs [5]. Anastomotic fungal infection or ulcerative tracheobronchitis usually occurs within 3 months of LTx, while invasive and disseminated fungal disease commonly develops during the first 6–12 months [1]. Fungal infection is related to high mortality and morbidity [6]. The 3-month mortality rate was approximately 22% in LTRs with IFIs, whereas the 1-year mortality was approximately 44% in a single-center study [7]. The incidence of invasive aspergillosis is exceptionally high post-LTx because the transplanted lungs bear the risk of constant exposure to the mold [1]. Additionally, the bronchial anastomotic site is susceptible to ischemic injury, necrosis, and potential infection with *Aspergillus* spp [8]. Infection with *Aspergillus* spp. is particularly critical, as it increases the risk of obstructive chronic lung allograft dysfunction (also known as obliterative bronchiolitis), the most common overall cause of death for patients after LTx [9].

A recent meta-analysis suggested that universal antifungal prophylaxis reduced the incidence of invasive aspergillosis after LTx [10]. However, no consensus has been reached regarding the initiation and duration of antifungal therapy [11], and transplant-centers employ different regimens of antifungal agents (AFAs) for prophylactic or preemptive purposes [12,13].

Therefore, the objectives of this study were to: (1) assess the epidemiology of fungus-positive respiratory samples (FPRSs), (2) describe the risk and prognostic factors of fungal infection after LTx, and (3) evaluate the effectiveness of our antifungal prophylaxis for LTx.

## 2. Materials and Methods

### 2.1. Study Design

This retrospective cohort study included all patients who received LTx between January 2012 and July 2019 at Severance Hospital. This hospital is a 2500-bed tertiary hospital where the first LTx was performed in South Korea, and over 40 lung transplant surgeries are still conducted every year. The registry had detailed medical records of LTRs and the clinical data of lung transplant donors. Patients with no signs of fungal infection after LTx were excluded. Of 242 cases, 132 cases of fungal infections were initially identified from the LTx registry (Figure 1). Among the suspected fungal infection cases, 10 recipients had evidence of pre-transplant aspergillosis detected in the explanted lungs, and five were fungus-positive in the implanted lungs. The species and isolation dates for FPRSs within the first year after LTx were identified, and some cases involved progression to invasive fungal diseases (IFDs). We also selected other episodes of IFDs included in the criteria.

This study protocol was approved by and followed the guidelines of the ethical review committee of Severance Hospital and the requirement for written informed consent was waived.

### 2.2. Data Collection

Culture, bronchoscopy, and laboratory findings were reviewed during episodes of infection. The type, duration, and modification of AFAs were included. The prognosis of IFDs, including treatment failure and mortality, was assessed. Using these data, cumulative incidence and mortality rates were calculated.

### 2.3. Lung Transplantation Protocol

In our institution, LTRs were treated according to the Yonsei Lung Transplantation Protocol established by a multidisciplinary team, which suggests a standardized guideline for administering immunosuppressants, antibiotics, and other medications. Initial immunosuppressive therapy after LTx included tacrolimus, methylprednisolone, and mycophenolate mofetil. Monoclonal antibodies such as basiliximab were only used in the vulnerable group who were at risk of subtherapeutic levels of calcineurin inhibitors. Initial empirical antibiotics included a combination of cefepime and teicoplanin in the perioperative period. Itraconazole (suspension solution 200 mg/20 mL twice a day) for universal antifungal prophylaxis and ganciclovir or valganciclovir for anti-cytomegalovirus prophylaxis was used for 6 months after surgery [14]. If patients received other AFAs before LTx, the therapy was either continued or adjusted depending on the cultured fungal genus.

In our institution, as itraconazole was used for prophylactic purposes during the initial 6 months after LTx, recipients with positive FPRSs, who were not included in the definition of probable or proven IFDs, did not receive augmented antifungal prophylaxis. However, if patients showed any signs suggestive of tracheobronchitis or invasive infection (candidemia or pneumonia), they were treated with other AFAs, usually echinocandin for *Candida* spp. and voriconazole for *Aspergillus* spp. Antimicrobial therapies were adopted according to the medical status of each patient.

Since azole AFAs inhibit the metabolism of tacrolimus, which is significantly metabolized by CYP3A4, tacrolimus levels and dose adjustment are necessary. In our institution, the tacrolimus trough level was tested daily during hospitalization, and the target levels were 8 to 10 mg/mL.

Donor airway samples were collected before implantation. Our team performed bronchoscopy regularly on schedule after LTx. Bronchoscopy was performed immediately after surgery, and at 2 and 4 weeks postoperatively; transbronchial biopsies and bronchoalveolar lavage (BAL) cultures were performed at 1, 3, 6, and 12 months after transplantation. Invasive respiratory tract samples were collected using bronchial washing or BAL for suspected infection. Serum and BAL galactomannan antigen levels were checked when fungal infection was suspected.

### 2.4. Definition

An FPRS was defined as any fungus-positive culture from the respiratory tract (sputum, BAL, bronchial washing, or biopsy). The definition of IFD, following the International Society for Heart and Lung Transplantation guideline, includes an FPRS in the presence of symptoms, radiological, and endobronchial changes, or the presence of histological changes consistent with fungal tissue invasion [11]. Invasive pulmonary mold disease was defined by the European Organization for Research and Treatment of Cancer/Mycoses Study Group [13], and positive value in galactomannan assay confirmed *Aspergillus* infection, besides the culture. We defined tracheobronchitis in IFD cases by the presence of endobronchial changes, such as ischemic bronchitis, bronchial necrosis, or pseudomembranes in the bronchial tree [15]. Anastomotic fungal infections were suspected based on airway irregularities, extraluminal air on chest imaging, or the presence of a pseudomembrane on bronchoscopic inspection. The diagnosis was confirmed with anastomotic fungal cultures, stains, and biopsies [1]. The date of diagnosis was defined as the date of the first positive culture or findings indicative of fungal infection. A patient could have multiple episodes of fungal infections with subsequent episodes caused by different fungi. A breakthrough fungal infection was defined as any clinical sign or symptom of IFIs occurring during exposure to at least 5 days of antifungal drug treatment, including fungi outside the spectrum of activity of the AFA [16]. Antifungal treatment failure is considered when patients present with clinical progression of infection despite AFA [17]. Persistent, refractory, or relapsed fungal infections were considered treatment failures in this study. Disseminated fungal infections in LTRs were characterized by bloodstream infections or the involvement of multiple noncontiguous sites [18].

### 2.5. Statistical Analyses

Continuous variables were presented as means and standard deviations or as medians and interquartile ranges (IQRs). Categorical variables were presented as numbers and percentages. The between-group difference was assessed using a one-way analysis of variance. Non-parametrically distributed data were analyzed with the Kruskal–Wallis test for between-group comparisons. Variables were compared using the Pearson’s chi-squared test or Fisher’s exact method for categorical data and independent t-test or Mann–Whitney U test for continuous data. Variables with a *p*-value of ≤0.1 in the univariate analysis were subjected to multivariate logistic regression analysis. The Kaplan–Meier method was used for the survival analysis, and the difference in survival was analyzed using the log-rank test. All categories were calculated as percentages with 95% confidence intervals (CIs); *p*-values <0.05 were considered statistically significant. Descriptive analyses were performed using R version 3.6.3 (R studio, Boston, MA, USA), while other analyses were performed using SPSS version 23.0 (IBM Corp., Armonk, NY, USA).

## 3. Results

### 3.1. Study Population

Among 132 patients with a suspected fungal infection, 102 (42.1%, 95% CI 35.9–48.4) presented FPRSs, and 90 (37.2%, 95% CI 31.1–43.3) developed IFDs within 1 year of LTx. The non-IFD group included 110 patients with no evidence of fungal infection, and 27 patients recovered from FPRSs without any sign of fungal infection (Figure 1). Nineteen patients had multiple episodes of FPRSs, and 10 experienced two episodes of IFDs within 1 year; 121 episodes of FPRSs and 99 episodes of IFDs were finally counted.

### 3.2. Fungus-Positive Respiratory Samples

*Candida* spp. and *Aspergillus* spp. were cultured from 85 (35.1%) and 29 (12.0%) patient samples at a median of 31.5 and 111.4 days after LTx, respectively. The cumulative incidence rates stratified by genus are presented in Figure 2. The other fungal genera cultured were *Rhizopus* spp., *Saccharomyces* spp., *Penicillium* spp., and *Blastochizomyces* spp. FPRSs were predominant in the first 90 days and the overall cumulative incidence of *Candida* spp. was approximately three times higher than that of *Aspergillus* spp. In the setting of itraconazole administration for 6 months post-LTx, *C. glabrata* accounted for 36.5% of all *Candida*-positive respiratory samples, followed by *C. albicans* (21.2%) and *C. tropicalis* (11.8%) (Table 1). Thirteen of twenty-nine (44.8%) *Aspergillus* spp. cases involved *A. fumigatus*; in the other cases, the strains were not identified.

### 3.3. Characteristics of Patients with Invasive Fungal Diseases

The baseline and perioperative characteristics of patients with IFDs (*n* = 90) and with no fungal infections (*n* = 137) after LTx are described in Table 2. The mean age of patients with IFDs was 55.8 ± 11.8 years, and 62.2% were men. Idiopathic pulmonary fibrosis was the highest causative lung disease of LTx (47.8%), followed by connective tissue disease-associated interstitial lung disease (CTD-ILD) (20%). One patient had previous kidney transplantation, and six had previous stem cell transplantations owing to hematologic malignancy. Five patients in the IFD group received single LTx. The univariate analysis of the IFD and non-IFD groups indicated that age and CTD-ILD showed significant differences between the groups (*p* = 0.019 and <0.001, respectively). The previous history denotes the 3-month period before LTx. Immunosuppressants included azathioprine, mycophenolate mofetil, or a dose of prednisolone greater than 20 mg. AFAs were used before LTx for treating fungal infection or colonization. The number of patients who received AFAs and/or were hospitalized before LTx was significantly higher in the IFD than in the non-IFD group. The median volume of blood loss during surgery was larger in the IFD than in the non-IFD group, although not statistically significant. Intensive care unit (ICU) stay and hospitalization days after LTx were longer in the IFD than in the non-IFD group (*p* = 0.020 and <0.001, respectively). Post-surgery events during hospitalization included (1) acute renal failure in patients who needed postoperative hemodialysis for various reasons, (2) weaning failure in patients who required a tracheostomy or home ventilation owing to respiratory failure, and (3) reoperation possibly because of massive bleeding or rejection. Weaning failure alone was higher in the IFD than in the non-IFD group (*p* = 0.058).

Independent risk factors associated with developing IFD (Table 3) included age in the 10-year groups (odds ratio (OR) 1.38; 95% CI 1.07–1.79), CTD-ILD (OR 10.55; 95% CI 2.85–39.10), hospitalization days after LTx in the 30-day groups (OR 1.24; 95% CI 1.08–1.42), and the previous use of AFAs (OR 2.32; 95% CI 1.07–5.02).

### 3.4. Clinical Course and Prognosis of Fungal Infection

Among 99 IFD episodes, 43 presented with candidiasis: 40 aspergillosis, 13 mixed fungal infections owing to polymicrobial traits, and two *Penicillium* spp. infection; additionally, one unclassified due to diagnostic limitation (Figure 3). The median duration from the date of transplant surgery to the date of the first IFD was 33 (IQR 13.5–77.3) and 103 (IQR 30.3–156) days in the *Candida* and *Aspergillus* groups, respectively.

Of the 43 candidiasis cases, tracheobronchitis accounted for 51.2% and disseminated and bloodstream infections for 34.9% and 14.0%, respectively. Disseminated cases showed fungal growth at multiple sites, particularly in urine, respiratory, and pleural effusion samples. Bronchoscopic findings of *Candida* tracheobronchitis indicated necrotic tissue, whitish plaque, ulcerative lesion, and gray balls. Three cases showed narrowing of the bronchus, and 11 cases had an anastomotic infection.

All 40 aspergillosis cases had invasive pulmonary aspergillosis, in which 17 were proven and 23 were probable. Twenty-six cases with tracheobronchitis showed white plaque, granulation tissue, and necrotic material. Other findings included anastomosis infection (five cases), bronchus stenosis (five cases), stent obstruction (one case), stent migration (one case), bronchial bleeding (one case), and bronchopleural fistula (one case). The mean value of the BAL galactomannan antigen level was 3.9 ± 2.36. The mixed fungal infection group (all cases) presented with tracheobronchitis, and the mean BAL galactomannan antigen level was 2.5 ± 2.07 (Figure 3).

The 1-year survival analysis did not indicate a statistical difference, although the IFD group had lower overall survival outcomes than the non-IFD group (log-rank *p* = 0.015, Figure 4). Among the 99 IFD cases, the treatment failure rate was 52.5% (52/99). The 1-year and overall mortality rates were 43.4% (43/99) and 58.6% (58/99) for the IFD group, respectively.

We performed a univariate analysis of overall survivors and non-survivors among IFD cases (Table 4). The time from surgery to an episode, postoperative complications, and fungal genus did not predict survival outcomes. However, fungal infection treatment failure led to a poor prognosis in terms of mortality. Some patients developed breakthrough fungal infections under treatment or prophylaxis for fungal infections. Seventeen patients were administered itraconazole before the breakthrough *Candida* infection. Among candidiasis patients, 25 showed treatment failure, and two succumbed to invasive candidiasis. *C. glabrata* was the predominant pathogen in these cases, which is mostly resistant to fluconazole and occasionally resistant to voriconazole. Even after the use of echinocandin, voriconazole, and polyene, four cases of *C. glabrata* and one each of *C. parapsilosis*, *C. tropicalis*, and *C. auris* infection showed refractory or relapsed IFDs. Host factors attributable to fungal infections, such as prolonged extracorporeal membrane oxygenation, chemotherapy due to hemophagocytic lymphohistiocytosis, multidrug-resistant bacterial infection worsened the host immune system and led to the fungal infection persisting. Among 20 episodes of treatment failure for aspergillosis, 18 involved breakthrough fungal infections: 14 were treated with itraconazole, three with voriconazole, and one with fluconazole. Five patients had conditions refractory to voriconazole, and their regimens were changed. However, three died from aspergillosis. Misdiagnosis of non-*Aspergillus* molds or *Aspergillus* spp. which are resistant to voriconazole could contribute to treatment failure.

## 4. Discussion

In our study, respiratory colonization of *Aspergillus* spp. and *Candida* spp. was noted in 12.0% and 35.1% of LTRs in the first year; three values were lower than those reported in a previous study [6]. The low rates may be attributed to the use of itraconazole, an AFA with broad-spectrum antifungal activity against yeasts and most molds. Regardless of fungal genus and prophylaxis, FPRSs were dominant in the first 3 months after LTx, possibly owing to the frequent collection of samples through bronchoscopy in the postoperative period (collection bias). Nevertheless, longer ICU stay, use of antibiotics and catheters could explain the colonization in LTRs. In addition, fungi that are less susceptible to itraconazole, such as *C. glabrata*, could survive and portend a high risk of subsequent invasive infection derived from tracheobronchitis, a unique entity and predominant manifestation in LTRs [19]. In particular, transplanted lungs are susceptible to anastomotic fungal infections in the early post-transplant period [1]. In our study, 19 patients showed anastomotic fungal infection, although discriminating fungal species was difficult through bronchoscopic findings. Fungal pseudomembranes occur in approximately half of the early bronchoscopies after LTx [20]. Therefore, it is crucial to inspect carefully and to suspect disease in the early post-LTx period.

Established variables associated with IFDs in LTRs were diabetes mellitus, low functional status, primary graft failure, extracorporeal membrane oxygenation, ventilator use, hemodialysis requirement in the perioperative period [21], and idiopathic pulmonary fibrosis [22]. Factors associated with invasive aspergillosis development were single LTRs and *Aspergillus* colonization at 1 year post-transplant [23]. In our study, old age, underlying CTD-ILD, and previous use of AFAs were predisposing factors for IFD development. CTD-ILD is a unique associative factor identified in our study. In our institution, systemic sclerosis and dermatomyositis are common underlying autoimmune diseases leading to CTD-ILD that are difficult to treat and that progress to end-stage lung disease despite active immunosuppressive therapy. The use of immunosuppressants before LTx and immune-mediated pathogenesis of the diseases could result in the development of IFD after LTx, which should be addressed in further research.

Overall, 37.2% (95% CI 31.1–43.3) of LTRs developed IFDs within the first year, although comparing IFD incidence rates across transplant centers is difficult owing to the different settings of immunosuppression and prophylaxis, and environmental epidemiology of fungi. The relatively high incidence rate of fungal infection in our institution could be because: (1) although itraconazole provides broad-spectrum antifungal activity, the agent has unpredictable pharmacokinetics [1], resulting in subtherapeutic drug serum levels and eventually breakthrough infections; (2) the incidence could be overestimated because attending physicians broadly included suspected fungal infections under IFDs to administer voriconazole (as required for medical coverage in South Korea), and itraconazole rather than voriconazole was used as the prophylactic AFA because of its low cost; and (3) multidrug-resistant pathogens such as *C. auris* have recently emerged and made treatment difficult, and they are threatening to health care settings [24].

The widespread use of azole antifungals has been associated with the emergence of resistant species and, particularly, a rise in multidrug-resistant *C. glabrata* in recent years [25]. Multidrug-resistant *C. glabrata* has been significantly associated with prior fluconazole or extensive echinocandin use [26]. In our study, nine episodes of IFDs caused by *C. glabrata* involved resistance to voriconazole after the use of itraconazole or voriconazole. The resistance of *C. glabrata* to caspofungin was not observed in our study. Therefore, it is highly recommended to use appropriate AFAs for isolated strains. To prevent the development of multidrug-resistant infections, AFAs should not be administered in subtherapeutic doses and therapy should not be continued for durations longer than indicated. Our study found statistically high mortality rates in the treatment failure fungal infection group. Therefore, in the high-risk fungal infection group, diligent efforts on proper mycological diagnosis and treatment with drug susceptibility tests are essential.

The primary limitations of our study are the single-center data collection and selection bias owing to its retrospective nature. Our findings may not be generalized to lung transplant centers with differences in fungal epidemiology or perioperative care. In addition, the low accuracy of identification and lack of drug sensitivity testing for mold strains could lead to misdiagnosed non-*Aspergillus* molds or contribute to treatment failure for *Aspergillus* spp., which is resistant to voriconazole.

However, to the best of our knowledge, this is the first study in South Korea describing the risk factors and clinical course of fungal infections during the first year after LTx. Our study identified several risk factors for the development of IFDs and mortality in the fungal infection group, which could contribute to the development of treatment strategies for IFDs in LTRs.

## Figures and Tables

**Figure 1 jof-07-00639-f001:**
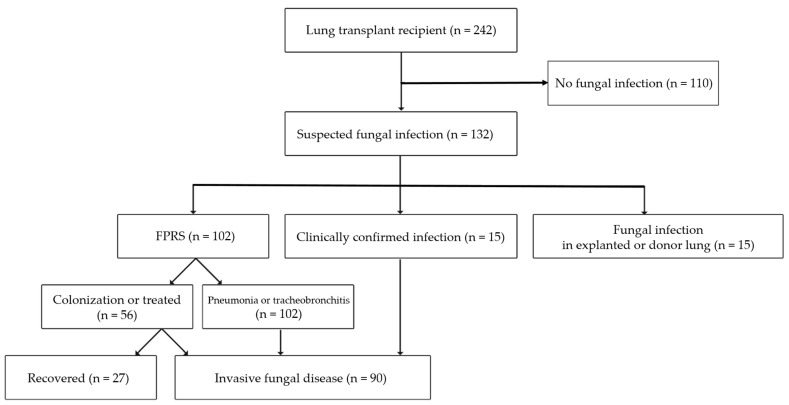
Summary of the study design. FPRS, fungus-positive respiratory sample.

**Figure 2 jof-07-00639-f002:**
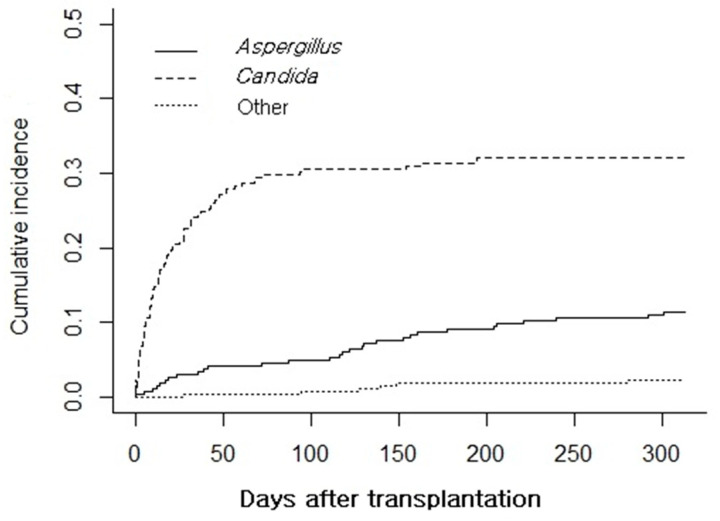
Cumulative incidence of fungus-positive respiratory samples stratified by fungal genus. The graph represents fungal isolation from the respiratory tract during antifungal prophylaxis.

**Figure 3 jof-07-00639-f003:**
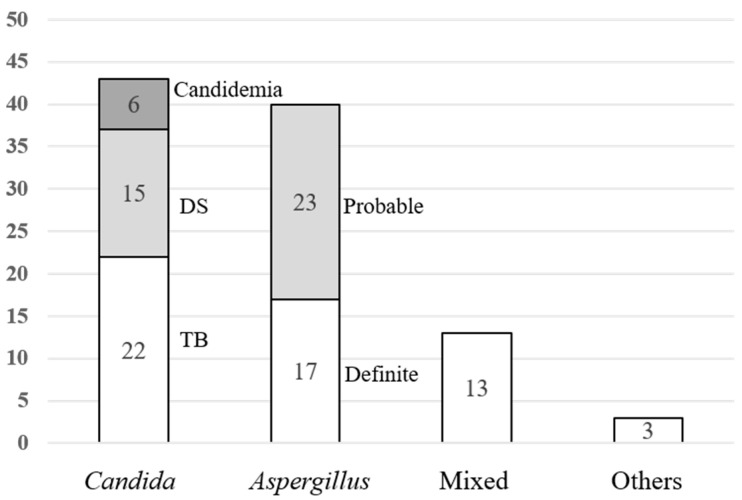
Classification of invasive fungal diseases. DS, disseminated fungal infection; TB, tracheobronchitis.

**Figure 4 jof-07-00639-f004:**
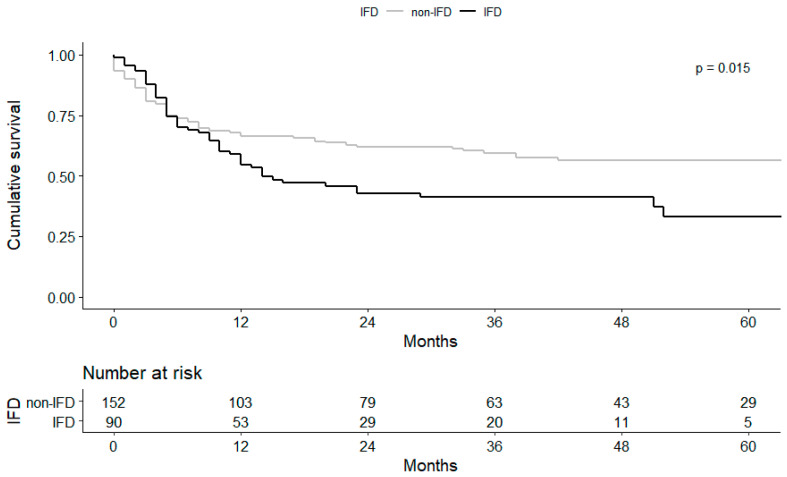
Association between fungal infection and mortality for the IFD and non-IFD groups.

**Table 1 jof-07-00639-t001:** *Candida* strains cultured from respiratory samples.

*Candida* Strains	Cases (*n* = 85)
*C. albicans*	18 (21.2)
Non-*albicans* strains	
*C. glabrata* ^1^	31 (36.5)
*C. tropicalis*	10 (11.8)
*C. parapsilosis*	6 (7.1)
*C. guilliermondii* ^2^	4 (4.7)
*C. lusitaniae* ^3^	1 (1.2)
*C. inconspicua* ^4^	1 (1.2)
*C. auris*	1 (1.2)
Unknown (yeast)	13 (15.3)

Data are presented as numbers (%). ^1^ Also known as *Nakaseomyces glabrata*. ^2^ Heterotypic synonym, current taxonomy is *Meyerozyma guilliermondii*. ^3^ Heterotypic synonym, current taxonomy is *Clavispora lusitaniae*. ^4^ Also known as *Pichia inconspicua*.

**Table 2 jof-07-00639-t002:** Baseline and perioperative characteristics of the study population.

Variable	IFDs (*n* = 90)	Non-IFDs (*n* = 137)	*p*-Value
Age, years	55.8 ± 11.8	51.8 ± 12.9	0.019
Sex, male	56 (62.2)	88 (64.2)	0.779
BMI	21.21 ± 4.2	20.86 ± 3.9	0.521
Disease leading to LTx			
Idiopathic pulmonary fibrosis	43 (47.8)	80 (58.4)	0.135
CTD-ILD	18 (20.0)	3 (2.3)	<0.001
Pulmonary tuberculosis	15 (16.7)	30 (21.9)	0.396
COPD	11 (12.2)	13 (9.5)	0.660
Other ILD	9 (10.0)	19 (13.9)	0.648
Obliterative bronchiolitis	7 (7.8)	13 (9.5)	0.812
Emphysema & bronchiectasis	6 (6.7)	9 (6.6)	0.977
Pulmonary hypertension	6 (6.7)	11 (8.0)	0.800
Lymphangioleiomyomatosis	2 (2.2)	2 (1.5)	0.65
Acute interstitial pneumonia	2 (2.2)	2 (1.5)	0.65
ARDS	1 (1.1)	6 (4.4)	0.249
Underlying disease			
DM	25 (27.8)	28 (25.7)	0.750
Connective tissue disease	22 (24.4)	17 (16)	0.155
Cancer	18 (20.0)	15 (14.4)	0.303
CAOD	10 (11.1)	6 (13.3)	0.780
Previous SCT	6 (6.7)	11 (8.1)	0.800
Previous SOT	1 (1.1)	6 (4.4)	0.483
Previous history within 3 months			
Hospitalization, yes	68 (75.6)	94 (68.6)	0.295
Hospitalization days before LTx	8 [0–33]	0 [0–16]	0.011
Antibiotics use	55 (61.1)	80 (58.4)	0.782
Antifungal agent use ^1^	25 (27.8)	19 (13.9)	0.011
Immunosuppressant use ^2^	52 (57.8)	77 (56.2)	0.891
Prior ECMO use	32 (35.6)	34 (24.8)	0.100
Transplantation type			
Single LTx	5 (5.4)	6 (4.4)	0.757
Surgery			
Operation time, minutes	412 [362–453]	392 [356–455]	0.480
Blood loss, L	2.45 [1.4–4.5]	2.00 [1.1–3.2]	0.057
Total ICU length of stay, days	19 [7–32]	11 [5–24]	0.020
Hospitalization days after LTx	62 [37.5–102.5]	35 [23.5–60.5]	<0.001
Post-surgery event in hospitalization period			
Acute renal failure ^3^	16 (17.8)	31 (22.6)	0.407
Weaning failure ^4^	54 (60.0)	64 (46.7)	0.058
Reoperation ^5^	26 (28.9)	27 (19.7)	0.148

The data are expressed as means ±SDs or medians (interquartile ranges) or numbers of patients (%). BMI, body mass index; CTD, connective tissue disease; ILD, interstitial lung disease; LTx, lung transplantation; COPD, chronic obstructive pulmonary disease; ARDS, acute respiratory distress syndrome; CAOD, coronary artery occlusive disease; SOT, solid organ transplantation; SCT, stem cell transplantation; ICU, intensive care unit; ECMO, extracorporeal membrane oxygenation; SD, standard deviation; DM, diabetes mellitus; IFD, invasive fungal disease. ^1^ Antifungal agents (AFAs) were used before LTx because of fungal infection or colonization. ^2^ Immunosuppressants included azathioprine, mycophenolate mofetil, or a dose of prednisolone greater than 20 mg. ^3^ Requiring post-operative hemodialysis for any reason. ^4^ Requiring a tracheostomy or home ventilator owing to respiratory failure. ^5^ Possibly due to massive bleeding or rejection.

**Table 3 jof-07-00639-t003:** Multivariate analysis of patients who developed fungal infections after lung transplantation.

Variable	Adjusted Odds Ratio (95% CI)	*p*-Value
Previous use of AFA	2.32 (1.07–5.02)	0.033
Age (by 10-year)	1.38 (1.07–1.79)	0.014
hospitalization days after LTx (by 30 days)	1.24 (1.08–1.42)	0.002
CTD-ILD	10.55 (2.85–39.10)	<0.001

CI, confidence interval; CTD-ILD, connective tissue disease-associated interstitial lung disease; AFA, antifungal agents.

**Table 4 jof-07-00639-t004:** Comparison of risk factors between overall survivors and non-survivors in the invasive fungal disease group.

	Survivors (*n* = 41)	Non-Survivors (*n* = 58)	*p*-Value
Time from surgery to an episode (d)	37 [21.5–133]	55.5 [19.5–131.3]	0.402
Post-surgery event in hospitalization period			
Acute renal failure ^1^	6 (35.3)	11 (64.7)	0.787
Weaning failure ^2^	22 (35.5)	40 (64.5)	0.143
Reoperation ^3^	11 (40.7)	16 (59.3)	1.000
Genus			
*Aspergillus* sp.	16 (40)	24 (60)	0.978
*Candida* sp.	17 (39.5)	26 (60.5)	0.899
Treatment failure	11 (21.2)	41 (78.8)	<0.001

The data are expressed as mean± SDs or medians (interquartile ranges) or numbers of patients (%). ^1^ Requiring post-operative hemodialysis for any reason. ^2^ Requiring a tracheostomy or home ventilator owing to respiratory failure. ^3^ Possibly due to massive bleeding or rejection.

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
