# Peer review of "The Prediction and Prognosis of Fungal Infection in Lung Transplant Recipients—A Retrospective Cohort Study in South Korea"

_jof, 2021, doi:10.3390/jof7080639_

Round 1
Reviewer 1 Report
The manuscript evaluates the risk factors and prognosis of fungal infection after lung transplantation. The manuscript is very interesting, well-structured and described in a clear and exhaustive way. The methodology is clearly and comprehensively described, and interpretation of the results well elucidated. Overall, the organization of the information is well presented. Specific comments: Line 82, 91, 131, 146 and 153: First line indent Line 153-154: How can we know the median from Figure 2? Line 157-158: as shown in figure 2, Candida-positive respiratory samples (CPRSs) were more predominant than Aspergillus-positive respiratory samples (APRSs) within 300 days, not only in the first 90 days. Line 159-161: what are the main species of Aspergillus. A figure of Candida spp and Aspergillus spp will be clearer and better. Line 162-165: please explain How did you do the fungal isolation, Did you sampling everyday or once a period of time? Line 211-229: A figure will be better and cleaner.Author Response
Point 1.Line 82, 91, 131, 146 and 153: First line indent Line
Response 1: I indented the lines.
Point 2. 153-154: How can we know the median from Figure 2?
Response 2: I modified the sentence: Candida spp. and Aspergillus spp. were cultured from 85 (35.1%) and 29 (12.0%) patients at a median of 31.5 and 111.4 days after LTx, respectively. The cumulative incidence stratified by genus is presented in Figure 2.
Point 3. Line 157-158: as shown in figure 2, Candida-positive respiratory samples (CPRSs) were more predominant than Aspergillus-positive respiratory samples (APRSs) within 300 days, not only in the first 90 days
Response 3: I changed the sentence: FPRSs were predominant in the first 90 days and the overall cumulative incidence of Candida spp. was approximately three times higher than that of Aspergillus spp.
Point 4. Line 159-161: what are the main species of Aspergillus. A figure of Candida spp and Aspergillus spp will be clearer and better.
Response 4: I added the sentence: Thirteen out of 29 (44.8%) of Aspergillus spp. were A. fumigatus, and others were not identified. And I also made the table of Candida strains
Point 5. Line 162-165: please explain How did you do the fungal isolation, Did you sampling every day or once a period of time?
Response 5: Additional protocol was written in 2.3 Lung transportation protocol as follows: Our team performed bronchoscopy regularly on schedule after LTx. Bronchoscopy was performed immediately after surgery and at 2, and 4 weeks postoperatively.
Point 6. Line 211-229: A figure will be better and cleaner.
Response 6: I added a new figure.
Reviewer 2 Report
This is a very interesting and important study. However, there seems to be a big problem.
- The following compares your data with Reference 6, but it is not appropriate to use only this content. There have been many reports in recent years, in particularly, I think your data is for candidiasis high mortality. The concern is that you have been investigating for about 7 years. I don't know the approval status of your country, I could not determine if the select of itraconazole was ethically valid.
- L.262. In our study, respiratory colonization of Aspergillus spp. and Candida spp. accounted for 12.0% and 35.1% of LTx in the first year, which was lower than that reported in a previous study—23% and up to 86%, respectively [6]. The low rate….
- The following is important information. L.160. C. glabrata accounted for 160 36.5% of all CPRSs, followed by C. albicans (21.2%) and C. tropicalis (11.8%).
- The important thing here is that fungal translocation seems to be the most suspicious at the time of C. glabrata. That is, problems with surgical procedures, excessive immunosuppression, etc. As you know, with Drug-Drug interaction, the azole increases the concentration of tacrolims.
- I have a question about the protocol. Why is the antibody agents, XXmab drug, not administered?
- L245. Information on breakthrough infections is very important. It is recommended to create a table because you can judge the appropriateness of treatment by describing the details in a table.
Author Response
Point 1. The following compares your data with Reference 6, but it is not appropriate to use only this content. There have been many reports in recent years, in particularly, I think your data is for candidiasis high mortality. The concern is that you have been investigating for about 7 years. I don't know the approval status of your country, I could not determine if the select of itraconazole was ethically valid.
Response 1: As I mentioned in the discussion, itraconazole is broad-spectrum antifungal activity against yeasts and most molds. In addition, it has less expensive and side effects than voriconazole. Therefore, we consider this agent has its own benefit despite a relatively high mortality rate that might result from different epidemiology and definition.
Point 2. The important thing here is that fungal translocation seems to be the most suspicious at the time of C. glabrata. That is, problems with surgical procedures, excessive immunosuppression, etc. As you know, with Drug-Drug interaction, the azole increases the concentration of tacrolims. I have a question about the protocol. Why is the antibody agents, XXmab drug, not administered?
Response 2: I added the contents about immunotherapies in 2.3 Lung transplantation protocol: Initial immunosuppressive therapy after LTx included tacrolimus, methylprednisolone, and mycophenolate mofetil. Monoclonal antibodies such as basiliximab were only used in the vulnerable group who were at risk of subtherapeutic levels of calcineurin inhibitors.
Point 3. L245. Information on breakthrough infections is very important. It is recommended to create a table because you can judge the appropriateness of treatment by describing the details in a table.
Response 3: I could not present the breakthrough infections in one table. Instead, I described the details (reasons) of the breakthrough infection in the paragraph.
Reviewer 3 Report
The paper by Yae Jee Baek is interesting but has the limitations of the single-center retrospective study. Abstract, Introduction and Discussion are concise and well written, the section of patient and methods needs to be improved. The references are updated.
The strength of the study relates to the number of patients registered in the study.
Issues:
1) immunosuppressive therapy must be explained more clearly (drug dose, duration, rejection episodes etc etc)
2) it is not indicated if the sensitivity of the fungi with respect to antifungal agents has been performed, particularly in cases of infection during prophylaxis
3) is not indicated if the study was approved by Ethics Committee or IRB
4) table 3 can be omitted and the results reported in the text 5) was the blood dosage of itraconazole comparable in the group of patients with fungal infection compared to those without infection?
Author Response
Point 1. Immunosuppressive therapy must be explained more clearly (drug dose, duration, rejection episodes etc etc)
Response 1: I add the contents of immunosuppressive therapy in 2.3 lung transplantation protocol: Initial immunosuppressive therapy after LTx included tacrolimus, methylprednisolone, and mycophenolate mofetil. Monoclonal antibodies such as basiliximab were only used in the vulnerable group who were at risk of subtherapeutic levels of calcineurin inhibitors
Point 2. It is not indicated if the sensitivity of the fungi with respect to antifungal agents has been performed, particularly in cases of infection during prophylaxis
Response 2: In the limitation part of the discussion, I wrote the low accuracy of identification and lack of drug sensitivity testing for mold strains. In terms of yeast, drug sensitivity tests were usually performed.
Point 3. Is not indicated if the study was approved by Ethics Committee or IRB
Response 3: I added the sentence in the end of 2.1 study design: This study protocol was approved by and followed the guidelines of the ethical review committee of Severance Hospital and the requirement for written informed consent was waived. Also, there are mentions of IRB statements in the footnote of the paper.
Point 4. Table 3 can be omitted and the results reported in the text
Response 4: I reported in the text, but to see the statistical significance, I left the table.
Point 5. Was the blood dosage of itraconazole comparable in the group of patients with fungal infection compared to those without infection?
Response 5: Itraconazole level is not routinely measured in our center. The subtherapeutic level might be a cause of the breakthrough infection which is described in the discussion.
Round 2
Reviewer 1 Report
The changes suggested by this reviewer were met and the authors provided the requested explanationsAuthor Response
Thanks for your comments.
I corrected minor spellings in the paper.
Reviewer 2 Report
The previous point 2 is not an answer. This is the basis of this research. Because it is necessary to describe the blood concentration of tacrolimus and mycophenolate mofetil. Also, I don't think there is a description of the ITCZ and methylprednisolone protocol. If you have high immunosuppression, you are more likely to get an infection. In this study, breakthrough infections and Candida infections are more frequent than in other studies.
Author Response
Thank you for your comments.
I read other articles and reviewed our study results.
My answer is as follows..
Itraconazole inhibits the metabolism of cyclosporine, tacrolimus, and sirolimus, which are significantly metabolized by CYP3A4. Therefore, tacrolimus levels and dose adjustment were required during itraconazole therapy.
In our center, 1mg of tacrolimus is administered right after surgery, and on the 1st day, 0.5mg was administered twice at 12-hour intervals, and the dosing was adjusted according to the tacrolimus trough level tested daily during hospitalization. The target tacrolimus trough levels were 8 to 10mg/ml. Mycophenolate mofetil was administered at 500mg twice a day, but the level of MMF was not performed in our institution. Methylprednisolone was administered at 500mg on surgery, followed by 0.5mg/kg divided by two for 3 days as induction therapy.
Interindividual variation of the cytochrome P450 genotypes which express phenotypically variable rates of metabolism and bioavailability due to different ingestion systems make variable pharmacokinetics of itraconazole. In other studies, serum itraconazole concentrations of >500 ng/mL are required to prevent Aspergillus species infection. Therefore, when patients were at high risk of systemic fungal infections or absorption or compliance were uncertain, trough levels are recommended. In our institution, high-risk patients had tested their serum itraconazole levels during anti-fungal prophylaxis, and some patients demonstrated the below of the required protective concentrations.
Therefore, even though immunosuppressants have effects on fungal infections in LTRs, the effect on high fungal infection rates was not expected to be significant in this study since the doses were not remarkably high.
I mentioned the hypotheses of relatively high fungal infection rates in the discussion (line 335); 1) subtherapeutic serum level of the itraconazole, 2) overestimation of invasive fungal infection due to medical coverage system in Korea, 3) the emergence (or spread) of multidrug-resistant fungi.
I wish this explanation could supplement our paper. In addition, this paper has been proofread by a professional editing company a couple of times. I hope the editing would meet the standard of JoF.
Thank you.
Reviewer 3 Report
The paper is now improveed in clarity, the authors full replied to criticisms
Author Response
Thank you very much for your comments and recommendations.